# Effect of Application Amounts on In Vitro Dermal Absorption Test Using Caffeine and Testosterone

**DOI:** 10.3390/pharmaceutics13050641

**Published:** 2021-04-30

**Authors:** Jueng-Eun Im, Hyang Yeon Kim, Jung Dae Lee, Jin-Ju Park, Kyung-Soo Kang, Kyu-Bong Kim

**Affiliations:** 1College of Pharmacy, Dankook University, 119 Dandae-ro, Cheonan 31116, Korea; 72180454@dankook.ac.kr (J.-E.I.); festivalkim@naver.com (H.Y.K.); ljd0734@nate.com (J.D.L.); pearl6230@naver.com (J.-J.P.); 2Toxicological Evaluation and Research Department, National Institute of Food and Drug Safety Evaluation, Ministry of Food and Drug Safety, Cheongju 28159, Korea; 3APURES Co., Ltd., Pyeongtae 13174, Korea; free8055@hanmail.net; 4Department of Animal Sciences, Shingu College, Gwangmyeong-ro 377, Jungwon-gu, Seongnam-si 13174, Korea

**Keywords:** in vitro, dermal absorption, caffeine, testosterone, application amounts

## Abstract

Dermal absorption of chemicals is a key factor in risk assessment. This study investigated the effects of different amounts of application on dermal absorption and suggested an appropriate application dose for proper dermal absorption. Caffeine and testosterone were chosen as test compounds. An in vitro dermal absorption test was performed using a Franz diffusion cell. Different amounts (5, 10, 25, and 50 mg (or µL)/cm^2^) of semisolid (cream) and liquid (solution) formulations containing 1% caffeine and 0.1% testosterone were applied to rat and minipig (Micropig^®^) skins. After 24 h, the concentrations of both compounds remaining on the skin surface and in the stratum corneum, dermis and epidermis, and receptor fluid were determined using LC-MS / MS or HPLC. Dermal absorption of both compounds decreased with increasing amounts of application in both skin types (rat and minipig) and formulations (cream and solution). Especially, dermal absorptions (%) of both compounds at 50 mg (or µL)/cm^2^ was significantly lower compared to 5 or 10 mg (or µL)/cm^2^ in both rat and minipig skins. Therefore, a low dose (5 or 10 mg (or µL)/cm^2^) of the formulation should be applied to obtain conservative dermal absorption.

## 1. Introduction

The dermal absorption rate is crucial for the risk assessment of cosmetics majorly absorbed through the skin [1,2,3,4,5,6,7,8] Several factors affect dermal absorption, such as donor characteristics, including gender [9], disease [10], age [11], race [12], vehicle [13], test substances (physicochemical properties, particle size) [14,15], skin condition [16], hydration [17], pH [18], stress [19], and physical or chemical damage [20], etc.

In vivo methods of determining the dermal absorption rate of cosmetic ingredients are not allowed anymore by regulatory agencies; only in vitro methods are. However, application dosages differ according to regulatory agencies. This study investigated the effects of different application doses using in vitro dermal absorption tests.

This study was performed using caffeine and testosterone as test materials. These were used as reference chemicals for skin absorption tests recommended by OECD TG 428. These chemicals had different physicochemical properties; LogKow values for caffeine and testosterone are −0.07 and 3.32, respectively, and molecular weights are 194.19 and 288.4 g/mol, respectively. In addition, these chemicals had a variety of skin absorption data [21,22,23,24,25,26,27,28,29,30,31,32].

Dermal absorption tests of caffeine and testosterone have been performed with different amounts of formulations, even with the same test substance. For caffeine, the applied dose range for liquid formulations was 10–764.2 μL/cm^2^ and for solid formulations 10–895.5 mg/cm^2^ [22,24,25,29,30,31,32] For testosterone, the applied dose range for liquid formulations was 10–400 μL/cm^2^ and for solid formulations 1.4–25 mg/cm^2^ [21,23,24,27,32].

However, the criteria for the dosage of formulations applied are unclear and impractical. The Organisation for Economic Co-operation and Development [33] and the Scientific Committee on Consumer Safety [34] recommend doses of 1–5 and 2–5 mg/cm^2^ for solid formulations of caffeine and testosterone, respectively, and up to 10 µL/cm^2^ for liquid formulations. In addition, there is a lack of studies on whether a different dosage can lead to a different dermal absorption rate. Therefore, this study determined pertinent application amounts for dermal absorption of caffeine and testosterone using an in vitro Franz diffusion cell. In addition, dermal absorptions of different formulations (cream and solution) with different animal skins (rat and minipig [Micropig^®^]) were compared. Different doses (5, 10, 25, and 50 mg/cm^2^ or μL/cm^2^), including doses given in OECD Guideline 428 [35], were tested. Two test compounds (caffeine and testosterone), which have different dermal absorption rates and different lipophilicities, were used; LogKow (Kow: partition coefficient between octanol and water) values for caffeine and testosterone are –0.07 and 3.32, respectively, and molecular weights are 194.19 and 288.4 g/mol, respectively (Table 1).

## 2. Materials and Methods

### 2.1. Chemicals

Caffeine and phosphate-buffered saline (PBS) as a receptor fluid (RF) were purchased from Sigma-Aldrich (product no. 27602; St. Louis, MO, USA), and testosterone (>98.0%), 4’-hydroxyacetanilide as an internal standard for caffeine analysis, and polyethylene glycol (PEG) monooleyl ether as a surfactant in the RF for the dermal absorption test of testosterone were purchased from Tokyo Chemical Industry Co., Ltd. (Tokyo, Japan). High-performance liquid chromatography (HPLC)-grade acetonitrile (ACN) and distilled water (DW) as the mobile phase for caffeine and testosterone analysis were obtained from Honeywell Burdick & Jackson Co (St. Harvey, MI, USA).

### 2.2. Formulations

The solutions were made of distilled water for 1% caffeine and 50% ethanol for 0.1% testosterone.

The cream formulations were made by Cosmecca Korea Inc. (Eumsung, Korea). It contained 1% caffeine or 0.1% testosterone. These were made using a mixture of several chemicals (sorbitan stearate, PEG-100 stearate, glyceryl stearate, cetearyl alcohol, polysorbate 60, hydrogenated polydecene, caprylic/capric triglyceride, dimethicone, disodium EDTA, chlorphenesin, glycerin, propanediol, distilled water, ammonium acryloyldimethyltaurate/VP copolymer, polyisobutene, polysorbate 20, polyacrylate-13, sorbitan isostearate, phenoxyethanol, caprylyl glycol and ethylhexylglycerin).

### 2.3. Skin Preparation

Rat and minipig skin membranes were prepared for this study. This study was approved by the Institutional Animal Care and Use Committee of Dankook University, South Korea (approval no. DKU-18-014). Male Sprague–Dawley (S-D) rats (age 7 weeks; body weight 228 ± 7 g) were purchased from Samtako Co. (Osan, Korea), and it took 1 week for domestication. The rats were kept in a specific pathogen-free (SPF) state in plastic cages under a strict light cycle at a temperature of 23 ± 2 °C with humidity of 50 ± 10% for 1 week. Standard feed (Samtako Co., Osan, Korea) and tap water were provided ad libitum. After 1 week, the rats were sacrificed using CO_2_ gas. Their dorsal hairs were shaved, and 7 × 7 cm^2^ sections of the shaved skin were cut. Next, 3 × 3 cm^2^ of full-thickness dorsal skin was loaded into a Franz diffusion cell. Minipig (Micropig^®^) skins were supplied from APURES Co., (Pyeongteak, Korea). The Micropig^®^ was raised in SPF zone at the temperature of 22 ± 2 °C and humidity of 50 ± 10%. Dorsal skin membranes were obtained from 6-month-old minipigs and standardized for dermal absorption experiments (thickness 500 μm; size 3 × 3 cm^2^). Each skin membrane was stored frozen at −20 °C until further use.

### 2.4. In Vitro Franz Diffusion Cell Equipment

All rat and Micropig^®^ skin membranes were mounted dermal side down in a Franz diffusion cell, which included a Vision Microette autosampler, circulating water bath, stirring plate, stirring control, and autofill (Hanson, Chatsworth, CA, USA). The volume of the RF chamber was 12 mL, the water bath was set to a temperature of 32.5 °C, and water was circulated to maintain a constant temperature of the RF. The receptor chamber was filled with PBS (pH 7.4) and 6% (*v*/*v*) PEG in PBS as an RF for caffeine and testosterone, respectively. Skin membranes were thawed to room temperature, loaded onto the Franz diffusion cell, and hydrated using normal saline. Prior to use, the integrity of skin membranes was assessed by measuring their transepidermal water loss (TEWL) using AquaFlux™ (AF200, Biox, London, UK). A skin area of 1.77 cm^2^ was exposed, and 5, 10, 25, and 50 mg/cm^2^ or 5, 10, 25, and 50 μL/cm^2^ of cream or liquid formulation, respectively, containing 1% caffeine or 0.1% testosterone was applied on the exposed area. When applied to this test area, the donor compartment was filled with approx. 8.85, 17.7, 44.25 and 88.5 mg (or µL) formulation using sterile syringes, in order to assure the appropriate test conditions. Only standard cell top components were fixed on the skin membranes to minimize back-diffusion during sampling. Next, 200 µL of RF samples were collected using an autosampler after rinsing it with fresh RF. The sampling time was set at 0, 1, 2, 4, 8, 12, and 24 h after application. The remaining formulation was rinsed off using alcohol swabs (BD™ Alcohol Swabs, Becton Dickinson, Franklin Lakes, NJ, USA) thrice, and the swabs were collected in an extracting solvent (40 mL of 50% [*v*/*v*] methanol for caffeine and 20 mL of methanol for testosterone). Throughout the process, WASH samples were obtained for each test. The residual formulation on stratum corneum (SC) samples was obtained by stripping the skin surface with Scotch tape (3M, Maplewood, MN, USA) 15 times. To obtain SKIN samples, the skin membranes used were cut into 8 pieces using surgical scissors and extracted in an appropriate solvent (20 mL of 50% [*v*/*v*] methanol for caffeine and 10 mL of methanol for testosterone). Total absorption was calculated as the ratio of caffeine or testosterone analyzed in SKIN and RF samples versus the total amount of caffeine or testosterone applied.

The study followed OECD Guideline 428 [35] and Guidance Document No. 28 [36].

### 2.5. Dermal Absorption Test of Caffeine

#### 2.5.1. Sample Preparation

After skin permeability testing for caffeine, all WASH, SC, and SKIN samples were extracted with 50% (*v*/*v*) methanol; the extract volume was 40 mL for WASH and 20 mL for SC and SKIN samples. RF samples contained in vials were transferred to Eppendorf tubes (Eppendorf, Hamburg, Germany), mixed using Multitube Vortexer (VX2500, VWR, Randor, PA, USA) for 10 min, and sonicated for 1 h. Each sample was stored at −4 °C for 24 h after sonication. For liquid chromatography–tandem mass spectrometry (LC-MS/MS) analysis, 50 μL of the samples in Eppendorf tubes were mixed with 400 μL of 50% (*v*/*v*) methanol containing 25 ng/mL of 4-hydroxyacetanilide (internal standard [IS]) using Multitube Vortexer for 30 s. All caffeine samples were centrifuged at 13,000 rpm for 10 min, and the supernatants were transferred to Norm-ject syringes (Henke Sass Wolf, Germany) and filtered using a 0.2 μm polytetrafluoroethylene (PTFE) filter (ADVANTEC, Dublin, CA, USA). Each sample was transferred to an autosampler vial (Thermo Fisher Scientific, Langerwehe, Germany), and 10 μL of each sample was injected into a column.

#### 2.5.2. Caffeine Analysis

Caffeine was analyzed using LC-MS/MS, and the analytical method was fully validated according to Ministry of Food and Drug Safety [37] and the US Food and Drug Administration [38] guidelines (Appendix A). The stability of caffeine was confirmed with various matrices. The final concentrations of calibration curve (CC) samples of caffeine were 0.02, 0.05, 0.1, 0.5, 0.7, 1, and 1.5 μg/mL. Validation points (quality control [QC] samples) included 0.05 (lower limit of quantitation [LLOQ]), 0.15, 0.6, and 1.2 μg/mL. Standard solutions (5 μL) (0.2, 0.5, 1, 5, 7, 10, and 15 μg/mL for CC samples and 0.5, 1.5, 6, and 12 μg/mL for QC samples) were diluted tenfold with 45 μL of a blank matrix (SKIN, WASH, SC, and R.F), and then 400 μL of 50% (*v*/*v*) methanol containing 25 ng/mL of 4-hydroxyacetanilide (IS) was added and mixed. All samples were filtered using a 0.2 μm PTFE filter before analysis.

Caffeine analysis was performed using Shimadzu 8040 LC-MS/MS (Shimadzu, Kyoto, Japan) equipped with an LC pump (LC-30AD-1 and LC-30AD-2), an autosampler (SIL 30AC), and a column oven (CTO-20AC). The analytical method was modified from Ref [39]. The system was interfaced through electrospray ionization (ESI) in the positive mode. Multiple reaction monitoring (MRM) was performed using the positive mode, and the MRM transition was set to m/s 195.10 → 138.10 for caffeine and 152.10 → 110.10 for 4-hydroxyacetanilide (IS). MRM was set up with the following parameters: ion source temperature in the desolvation line and heater block was 250 °C and 400 °C, respectively; nebulizing and drying gas flow was 2 L/min; dwell time was 100 s; collision-induced dissociation energy (CE) of caffeine was −20 eV and Q1 prebias and Q3 prebias were −11 and −13 V, respectively; and CE of 4-hydroxyacetanilide (IS) was −17 eV and Q1 prebias and Q3 prebias were −25 and −11 V, respectively. A thermo hypersil GOLD^TM^ phenyl (250 × 10 mm, 5 μm) (part no. 25903-153030) column with a guard column (Security Guard Cartridges RP-1, 4 × 3.0 mm; Phenomenex, CA, USA) was used for analysis. The column oven temperature was maintained at 40 °C. A gradient system of water and ACN was used for 6.5 min at a flow rate of 0.3 mL/min. ACN concentration was increased from 20% to 65% for 0–4 min, maintained at 65% for 4–5.5 min, then decreased to 20% for 5.5–6 min, and maintained at 20% for 6–6.5 min. The retention time of caffeine and 4-hydroxyacetanilide (IS) was 4.2 and 3.5 min, respectively.

### 2.6. Dermal Absorption Test of Testosterone

#### 2.6.1. Sample Preparation

After skin permeability testing for testosterone, all WASH, SC, and SKIN samples were extracted with methanol; the extract volume was 20 mL for WASH and 10 mL for SC and SKIN samples. RF samples contained in vials were transferred to Eppendorf tubes. The samples were mixed using a Multitube Vortexer for 10 min and then sonicated for 1 h. Each sample was stored at −4 °C for 24 h after sonication. For HPLC analysis, 200 μL of ACN was added to 50 μL of each sample and mixed using Multitube Vortexer for 30 s. All samples were then centrifuged at 13,000 rpm for 10 min, and the supernatants were transferred to Norm-ject syringes and filtered using a 0.2 μm PTFE filter. Finally, each sample was transferred to an autosampler vial (Thermo Fisher Scientific, Waltham, MA, USA), and 10 μL of each sample was injected into a column.

#### 2.6.2. Testosterone Analysis

Testosterone was analyzed using HPLC, and the analytical method was fully validated according to MFDS [37] and US FDA [38] guidelines (Appendix A). The stability of testosterone was confirmed with various matrices. The final concentrations of CC samples for testosterone were 0.1, 0.5, 0.8, 1, 5, and 10 μg/mL. Validation points (QC samples) included 0.1 (LLOQ), 0.3, 3, and 8 μg/mL. Standard solutions (5 μL) (1, 5, 8, 10, 50, and 100 μg/mL for CC samples and 1, 3, 30, and 80 μg/mL for QC samples) were diluted tenfold with 45 μL of a blank matrix (SKIN, WASH, SC, and RF) and then 200 μL ACN solution was added and mixed. All samples were filtered using a 0.2 μm PTFE filter before analysis.

HPLC analysis of testosterone was performed using Agilent 1290 infinity LC (Agilent Technology, Waldbronn, Germany). The analytical method was modified from Refs [40]. Samples were detected using an ultraviolet-visible (UV-Vis) spectroscopy detector, and the wavelength was set to 254 nm. A 250 × 4.6 mm E11070 KR100-5C18 column (Kromasil^®^, Bohus, Sweden) with a guard column (Security Guard Cartridges RP-1, 4 × 3.0 mm) was used for analysis. The temperature of the column and autosampler was maintained at 37 °C and 4 °C, respectively. RF, SC, and SKIN samples were analyzed using an isocratic system of 60% ACN and 40% DW, while WASH samples were analyzed using 50% ACN and 50% DW. The injection volume was 10 μL, and the flow rate was maintained at 1.5 mL/min.

### 2.7. Statistics

All data were presented as mean ± standard deviation (SD). Statistical analysis was performed using Microsoft Excel 2016 (Microsoft Corporation, Redmond, WA, USA) and GraphPad Prism version 5.01 (GraphPad, San Diego, CA, USA). Differences in skin permeability and dermal absorption at four different dosages were statistically tested using one-way analysis of variance (ANOVA) followed by Tukey’s test (*p* < 0.05). In addition, statistical differences between skin permeability and absorption in the two formulations (cream and liquid) were confirmed by Student’s *t*-test (*p* < 0.05).

## 3. Results

### 3.1. In Vitro Dermal Absorption Study of Caffeine

#### 3.1.1. Rat Skin

With 5 mg/cm^2^ of cream formulation, the caffeine content of SC, SKIN, and RF samples was 3.7% ± 2.0%, 33.9% ± 12.8%, and 12.2% ± 5.0%, respectively. The dermal absorption rate was 46.0% ± 13.3%, and total recovery was 92.5% ± 2.9%, including 42.8% ± 11.8% of caffeine in WASH samples (Figure 1A). With 10 mg/cm^2^ of cream formulation, the caffeine content of SC, SKIN, and RF samples was 8.7% ± 6.5%, 26.9% ± 8.6%, and 10.1% ± 7.4%, respectively. The dermal absorption rate was 37.0% ± 8.8%, and total recovery was 107.6% ± 6.6%, including 61.9% ± 14.5% of caffeine in WASH samples (Figure 1B). With 25 mg/cm^2^ of cream formulation, the caffeine content of SC, SKIN, and RF samples was 1.9% ± 1.0%, 31.5% ± 12.2%, and 4.8% ± 2.7%, respectively. The dermal absorption rate was 36.3% ± 14.2%, and total recovery was 109.8% ± 7.5%, including 71.7% ± 18.1% of caffeine in WASH samples (Figure 1C). With 50 mg/cm^2^ of cream formulation, the caffeine content of SC, SKIN, and RF samples was 2.0% ± 0.9%, 25.3% ± 9.3%, and 4.1% ± 2.0%, respectively. The dermal absorption rate was 29.4% ± 10.5%, and total recovery was 118.1% ± 4.7%, including 86.4% ± 10.6% of caffeine in WASH samples (Figure 1D).

With 5 μL/cm^2^ of liquid formulation, the caffeine content of SC, SKIN, and RF samples was 2.7% ± 1.7%, 13.4% ± 4.1%, and 23.4% ± 13.9%, respectively. The dermal absorption rate was 36.7% ± 11.2%, and total recovery was 92.9% ± 9.3%, including 53.5% ± 6.6% of caffeine in WASH samples (Figure 1A). With 10 μL/cm^2^ of liquid formulation, the caffeine content of SC, SKIN, and RF samples was 5.0% ± 1.6%, 36.1% ± 16.1%, and 12.5% ± 9.9%, respectively. The dermal absorption rate was 48.6% ± 14.4%, and total recovery was 100.9% ± 6.5%, including 47.3% ± 19.7% of caffeine in WASH samples (Figure 1B). With 25 μL/cm^2^ of liquid formulation, the caffeine content of SC, SKIN, and RF samples was 2.9% ± 2.3%, 28.0% ± 15.4%, and 11.9% ± 3.1%, respectively. The dermal absorption rate was 39.9% ± 16.8%, and total recovery was 97.7% ± 9.5%, including 55.0% ± 10.6% of caffeine in WASH samples (Figure 1C). With 50 μL/cm^2^ of liquid formulation, the caffeine content of SC, SKIN, and RF samples was 1.5% ± 0.3%, 31.4% ± 10.7%, and 9.1% ± 4.1%, respectively. The dermal absorption rate was 40.5% ± 11.6%, and total recovery was 83.8% ± 21.5%, including 41.8% ± 18.9% of caffeine in WASH samples (Figure 1D). All recoveries (%) met the acceptable range of 80–120% according to OECD guidelines [36] for nonradioactive labeling substances (Figure 1).

#### 3.1.2. Minipig Skin

With 5 mg/cm^2^ of cream formulation, the caffeine content of SC, SKIN, and RF samples was 5.7% ± 1.8%, 4.7% ± 0.4%, and 51.8% ± 14.1%, respectively. The dermal absorption rate (SKIN + RF samples) was 56.4% ± 14.2%, and total recovery was 92.9% ± 7.4%, including 30.8% ± 7.4% of caffeine in WASH samples (Figure 2A). With 10 mg/cm^2^ of cream formulation, the caffeine content of SC, SKIN, and RF samples was 3.4% ± 1.7%, 5.2% ± 1.3%, and 70.5% ± 3.1%, respectively. The dermal absorption rate was 75.7% ± 2.4%, and total recovery was 109.0% ± 7.7%, including 30.0% ± 8.3% of caffeine in WASH samples (Figure 2B). With 25 mg/cm^2^ of cream formulation, the caffeine content of SC, SKIN, and RF samples was 2.9% ± 0.4%, 6.3% ± 1.5%, and 14.5% ± 10.1%, respectively. The dermal absorption rate was 20.8% ± 10.5%, and total recovery was 90.3% ± 4.6%, including 66.6% ± 12.3% of caffeine in WASH samples (Figure 2C). With 50 mg/cm^2^ of cream formulation, the caffeine content of SC, SKIN, and RF samples was 1.8% ± 0.3%, 2.9% ± 1.0%, and 6.8% ± 2.1%, respectively. The dermal absorption rate was 9.7% ± 2.4%, and total recovery was 102.2% ± 8.1%, including 90.8% ± 7.8% of caffeine in WASH samples (Figure 2D).

With 5 μL/cm^2^ of liquid formulation, the caffeine content of SC, SKIN, and RF samples was 3.7% ± 3.2%, 6.5% ± 2.5%, and 84.3% ± 9.7%, respectively. The dermal absorption rate was 90.7% ± 10.7%, and total recovery was 116.7% ± 15.4%, including 22.3% ± 13.3% of caffeine in WASH samples (Figure 2A). With 10 μL/cm^2^ of liquid formulation, the caffeine content of SC, SKIN, and RF samples was 3.9% ± 3.0%, 5.9% ± 2.2%, and 54.4% ± 23.1%, respectively. The dermal absorption rate was 60.3% ± 21.3%, and total recovery was 102.6% ± 24.4%, including 38.3% ± 5.6% of caffeine in WASH samples (Figure 2B). With 25 μL/cm^2^ of liquid formulation, the caffeine content of SC, SKIN, and RF samples was 2.4% ± 2.3%, 4.5% ± 1.7%, and 20.3% ± 10.9%, respectively. The dermal absorption rate was 24.8% ± 9.9%, and total recovery was 99.8% ± 10.5%, including 72.6% ± 18.1% of caffeine in WASH samples (Figure 2C). With 50 μL/cm^2^ of liquid formulation, the caffeine content of SC, SKIN, and RF samples was 1.9% ± 1.1%, 3.2% ± 2.1%, and 13.3% ± 12.2%, respectively. The dermal absorption rate was 16.5% ± 10.6%, and total recovery was 99.3% ± 5.0%, including 81.0% ± 8.7% of caffeine in WASH samples (Figure 2D). All recoveries (%) met the acceptable range of 80–120% according to OECD guidelines [41] for nonradioactive labeling substances (Figure 2).

### 3.2. In Vitro Dermal Absorption Study of Testosterone

#### 3.2.1. Rat Skin

With 5 mg/cm^2^ of cream formulation, the testosterone content of SC, SKIN, and RF samples was 4.6% ± 3.2%, 23.7% ± 3.1%, and 18.0% ± 4.6%, respectively. The dermal absorption rate (SKIN + RF samples) was 41.7% ± 5.7%, and total recovery was 94.1% ± 10.9%, including 47.8% ± 8.8% of testosterone in WASH samples (Figure 3A). With 10 mg/cm^2^ of cream formulation, the testosterone content of SC, SKIN, and RF samples was 5.0% ± 2.1%, 22.9% ± 10.1%, and 10.7% ± 2.6%, respectively. The dermal absorption rate was 33.6% ± 9.9%, and total recovery was 94.2% ± 7.5%, including 55.6% ± 9.7% of testosterone in WASH samples (Figure 3B). With 25 mg/cm^2^ of cream formulation, the testosterone content of SC, SKIN, and RF samples was 8.4% ± 4.5%, 34.8% ± 5.1%, and 5.1% ± 1.5%, respectively. The dermal absorption rate was 39.9% ± 4.0%, and total recovery was 99.3% ± 2.2%, including 51.0% ± 2.8% of testosterone in WASH samples (Figure 3C). With 50 mg/cm^2^ of cream formulation, the testosterone content of SC, SKIN, and RF samples was 2.1% ± 1.3%, 12.3% ± 4.0%, and 1.9% ± 0.3%, respectively. The dermal absorption rate was 14.2% ± 4.0%, and total recovery was 81.6% ± 3.2%, including 65.4% ± 1.0% of testosterone in WASH samples (Figure 3D).

With 5 μL/cm^2^ of liquid formulation, the testosterone content of SC, SKIN, and RF samples was 12.2% ± 3.0%, 50.1% ± 15.5%, and 22.4% ± 2.3%, respectively. The dermal absorption rate was 72.4% ± 13.9%, and total recovery was 98.7% ± 7.7%, including 14.1% ± 10.5% of testosterone in WASH samples (Figure 3A). With 10 μL/cm^2^ of liquid formulation, the testosterone content of SC, SKIN, and RF samples was 3.3% ± 2.2%, 30.5% ± 6.0%, and 18.1% ± 4.0%, respectively. The dermal absorption rate was 48.6% ± 6.5%, and total recovery was 102.3% ± 8.0%, including 50.4% ± 13.8% of testosterone in WASH samples (Figure 3B). With 25 μL/cm^2^ of liquid formulation, the testosterone content of SC, SKIN, and RF samples was 4.4% ± 0.9%, 38.9% ± 6.5%, and 18.0% ± 3.1%, respectively. The dermal absorption rate was 56.8% ± 8.6%, and total recovery was 101.2% ± 5.1%, including 39.9% ± 10.4% of testosterone in WASH samples (Figure 3C). With 50 μL/cm^2^ of liquid formulation, the testosterone content of SC, SKIN, and RF samples was 3.4% ± 1.9%, 37.8% ± 4.6%, and 9.9% ± 3.7%, respectively. The dermal absorption rate was 47.6% ± 7.4%, and total recovery was 94.0% ± 3.4%, including 43.1% ± 10.9% of testosterone in WASH samples (Figure 3D). All recoveries (%) met the acceptable range of 80.5–120% according to OECD guidelines [41] for nonradioactive labeling substances (Figure 3).

#### 3.2.2. Minipig Skin

With 5 mg/cm^2^ of cream formulation, the testosterone content was below the LLOQ in SC samples 13.8% ± 4.7% and 20.4% ± 4.1% in SKIN and RF samples, respectively. The dermal absorption rate (SKIN + RF samples) was 34.2% ± 7.5%, and total recovery was 111.6% ± 15.2%, including 77.4% ± 20.6% of testosterone in WASH samples (Figure 4A). With 10 mg/cm^2^ of cream formulation, the testosterone content was below the LLOQ in SC samples and 12.0% ± 2.2% and 11.7% ± 4.1% in SKIN and RF samples, respectively. The dermal absorption rate was 23.7% ± 2.2%, and total recovery was 110.9% ± 4.1%, including 87.2% ± 2.8% of testosterone in WASH samples (Figure 4B). With 25 mg/cm^2^ of cream formulation, the testosterone content was below the LLOQ in SC samples and 24.5% ± 10.6% and 8.5% ± 3.2% in SKIN and RF samples, respectively. The dermal absorption rate was 33.0% ± 10.9%, and total recovery was 93.7% ± 3.9%, including 60.6% ± 8.3% of testosterone in WASH samples (Figure 4C). With 50 mg/cm^2^ of cream formulation, the testosterone content of SC, SKIN, and RF samples was 0.5% ± 1.0%, 6.9% ± 6.7%, and 6.3% ± 1.5%, respectively. The dermal absorption rate was 13.2% ± 5.2%, and total recovery was 80.8% ± 2.7%, including 67.1% ± 3.9% of testosterone in WASH samples (Figure 4D).

With 5 μL/cm^2^ of liquid formulation, the testosterone content was below the LLOQ in SC samples and 19.9% ± 10.6% and 47.8% ± 19.2% in SKIN and RF samples, respectively. The dermal absorption rate was 67.7% ± 20.6%, and total recovery was 97.0% ± 12.8%, including 29.3% ± 24.9% of testosterone in WASH samples (Figure 4A). With 10 μL/cm^2^ of liquid formulation, the testosterone content of SC, SKIN, and RF samples was 4.8% ± 4.0%, 13.1% ± 4.7%, and 45.7% ± 9.0%, respectively. The dermal absorption rate was 58.8% ± 10.6%, and total recovery was 89.9% ± 6.7%, including 26.3% ± 2.7% of testosterone in WASH samples (Figure 4B). With 25 μL/cm^2^ of liquid formulation, the testosterone content of SC, SKIN, and RF samples was 13.3% ± 9.8%, 31.2% ± 27.0%, and 15.3% ± 6.5%, respectively. The dermal absorption rate was 46.5% ± 24.9%, and total recovery was 92.7% ± 17.4%, including 32.9% ± 9.0% of testosterone in WASH samples (Figure 4C). With 50 μL/cm^2^ of liquid formulation, the testosterone content of SC, SKIN, and RF samples was 6.7% ± 6.3%, 16.8% ± 16.3%, and 7.5% ± 1.4%, respectively. The dermal absorption rate was 24.2% ± 17.4%, and total recovery was 83.1% ± 12.8%, including 52.2% ± 14.1% of testosterone in WASH samples (Figure 4D). All recoveries (%) met the acceptable range of 80–120% according to OECD guidelines [41] for nonradioactive labeling substances (Figure 4).

## 4. Discussion

The dermal absorption is an important parameter for calculating systemic exposure dosage (SED) in cosmetic risk assessment. However, skin absorption rate of the test substance could be changed depending on the test conditions. For this reason, standardization of test conditions is necessary. In this study, among the conditions of the skin absorption test, the effect of “application amount” on the skin absorption rate was evaluated. Then, the most practical application amount to obtain a conservative skin absorption rate was searched.

We used a semi-solid and liquid formulation for the test. The reason for using the semi-solid formulation was to simulate and evaluate the cosmetic formulation. The reason for choosing the liquid vehicle as an ethanol solution was to dissolve the test substance, so it was inevitable to use an organic solution, even if it could affect the skin condition. In addition, the ethanol aqueous solution had the advantage that it is easy to compare the results as it is used in various literatures.

The Micropig skin used in the test was 500 µm thickness, and the full skin of rat was used. This was used because the TEWL values (7.4 ± 0.5 g/h/m^2^, Micropig; 13.5 ± 1.3 g/h/m^2^, Rat) were most similar to men’s forearm global TEWL standard (7.7 ± 2 g/h/m^2^) as an indicator of the integrity of the skin in the test (data not shown) (https://info.kcii.re.kr/ (accessed on 30 April 2021)). The TEWL value is a factor that describes the properties of the skin barrier, and the skin barrier has a great influence on the rate of skin absorption [42]. To use the skin that best mimics the human skin barrier, the thickness condition for each skin was chosen based on similar TEWL values. In addition, the optimum thickness with stable TEWL value has been selected.

This study was performed using caffeine and testosterone as test materials. Several studies to determine the skin absorption rate of the reference chemicals (caffeine and testosterone) have been conducted under a wide range of application conditions, and their absorption rates vary from study to study. Previous dermal absorption studies on caffeine and testosterone with different formulation dosage conditions (10–895.5 mg/cm^2^ or 10–764.2 µL/cm^2^ for caffeine and 1.4–25 mg/cm^2^ or 10–400 µL/cm^2^ for testosterone) [21,22,23,24,25,26,27,28,29,30,31,32] showed different dermal absorption rates. The skin absorption amounts of testosterone were increased as the amount of application per unit area (or concentration) increased, but the percentages of skin absorption were similar or decreased as the amount of application per unit area increased [26].

In this study, it was also confirmed that applying an excessive amount to the skin caused saturation to the skin, and the skin absorption rate decreased. We observed an inverse relationship between dosage and dermal absorption (Figure 5). Therefore, it was confirmed that a small amount of application caused a conservative skin absorption rate.

So, it seemed to be reasonable that the dose to obtain skin absorption rate, known as a “finite” dose, was defined as a rather small dose in several international regulations. However, it needed to be standardized because the criteria for “finite dose” was various. The World Health Organization [43] and OECD Guideline 428 [35] suggest 1–5 mg/cm^2^ or 10 µL/cm^2^ as a finite dose, and OECD Guidance Document No. 28 suggests 10 mg/cm^2^ or 10 µL/cm^2^ [36]. SCCS [34] suggests 2–5 mg/cm^2^ or up to 10 µL/cm^2^ and up to 20 mg/cm^2^ for oxidative hair dyes as a finite dose.

For the standardization, it was necessary to enable practical and efficient testing. It is difficult to apply a small amount of test substance such as 1–5 mg/cm^2^ (or µL/cm^2^). It means that trace level amounts (less than about 4–10 mg or µL) should be applied to the narrowly exposed area (<2 cm^2^) when performing an in vitro skin absorption test on a Franz diffusion cell. This difficulty could cause considerable variation in tests, so it is also important to determine practical application conditions as they are large enough to be tested. Previous studies have also shown that experiments at low loads present technical complications for the investigator [44,45,46].

Accordingly, in this study, skin absorption tests were conducted under the practical conditions of applied amounts of 5, 10, 25, and 50 mg/cm^2^ (µL/cm^2^), and these are generally less than the “infinite dose” (>100 µL/cm^2^); OECD [36] described when infinite dose conditions of exposure are used the data may permit the calculation of a permeability constant (Kp), but the percentage absorbed is not relevant.

In the results, it showed dermal absorption with a dose of 5 and 10 mg/cm^2^ or 5 and 10 μL/cm^2^ was significantly different from that with a dose of 25 and 50 mg/cm^2^ or 25 and 50 μL/cm^2^. So, the ranges of 5 and 10 mg/cm^2^ or 5 and 10 μL/cm^2^ were reasonable.

In the caffeine tests using Micropig^®^ skin, dermal absorption was 56.4–68.1% in a cream formulation and 60.3–90.7% in a liquid formulation at a lower dose (5 and 10 mg/cm^2^ or 5 and 10 μL/cm^2^), and it significantly decreased to 10–20.8% in a cream formulation and 16.4–24.8% in a liquid formulation at a higher dose (25 and 50 mg/cm^2^ or 25 and 50 μL/cm^2^) (Figure 5B). Compared with the known human maximum caffeine absorption rate (47.6%) [27], it showed a high or similar skin absorption rate in the range of 5 and 10 μL/cm^2^. The semi-solid formulation (5 and 10 mg/cm^2^) has not been reported yet. Meanwhile, no statistically significant difference was found in the test results in the caffeine tests using rat skin (Figure 5A).

In the testosterone tests using Micropig^®^ skin, dermal absorption was 23.7–34.2% in a cream formulation and 58.8–67.7% in a liquid formulation at a lower dose (5 and 10 mg/cm^2^ or 5 and 10 μL/cm^2^), and it significantly decreased to 13.2–33.1% in a cream formulation and 24.2–46.5% in a liquid formulation at a higher dose (25 and 50 mg/cm^2^ or 25 and 50 μL/cm^2^) (Figure 5D). Compared with the known human maximum caffeine absorption rate (49.5%) [27], it showed a high or similar skin absorption rate in the range of 5 and 10 μL/cm^2^.

In the testosterone tests using rat skin, dermal absorption was 33.6–41.7% in a cream formulation and 48.6–72.4% in a liquid formulation at a lower dose (5 and 10 mg/cm^2^ or 5 and 10 μL/cm^2^), and it decreased to 14.2−16.8% in a cream formulation and 47.6–56.8% in a liquid formulation at a higher dose (25 and 50 mg/cm^2^ or 25 and 50 μL/cm^2^) (Figure 5C). Similar to previous test results, they showed higher or similar skin absorption rates in the range of 5 to 10 μL/cm^2^ compared to known human maximum caffeine absorption (49.5%) [27].

Accordingly, cream formulations showed the highest dermal absorption rate at 5 or 10 mg/cm^2^, and the dose was suitable to obtain conservative dermal absorption (Figure 5B). For liquid formulations, the highest dermal absorption rate was found at 5 or 10 μL/cm^2^, and dermal absorption at that dose appears to be the most appropriate for conservative SED values (Figure 5). We could also show that this dose range is suitable for obtaining dermal absorption rate based on the kinetics of penetration. In most of the penetration kinetics graphs (Figure 6 and Figure 7) for the two test substances the slope of the penetration graph gradually decreased in the test under the application dose condition of 5 and 10 mg/cm^2^ or 25 and 50 μL/cm^2^, and it meant that these doses were the “finite dose” (condition used to calculate the dermal absorption rate).

However, some exceptions appeared in the case using Micropig skin and a cream formulation—the slope of penetration graph (the slope of the graph) tended to increase gradually (Figure 6C and Figure 7C). Therefore, it was difficult to define 5 and 10 mg/cm^2^ as a “finite dose” under this condition.

Nevertheless, it showed a sufficiently high dermal absorption rate at 5 and 10 mg/cm^2^, and as mentioned above, the dermal absorption rate was similar to that of the human in vivo dermal absorption rate. Therefore, it was judged to be still suitable for the test.

In conclusion, a dose of 5 or 10 mg/cm^2^ or 5 or 10 μL/cm^2^ in cream or liquid formulation, respectively, is recommended to obtain conservative dermal absorption of a substance for risk exposure assessment. It was similar or relaxed to the dose range of the existing regulations [34,35,36,43]. The proposed dose range is also within the 0.005,714–18.75 mg/cm^2^ daily dose per surface area range for each cosmetic product in Korea [47] and 0.01086–15.63 mg/cm^2^ in Europe [34] on the basis of statistical data of cosmetic use per body surface area. So, the selected application condition seems to be practical.

This study was a skin absorption test method using Franz diffusion cells, which is applicable to cosmetics research as an animal replacement test method. The method using minipigs (Micropig^®^), which can be used for human organ transplantation and the remaining skin as an experimental donor, is superior to the in vitro test method using rat skin in terms of animal ethics.

It was expected that other drugs will have a similar trend to the results of this study. In general, the dermal absorption rate is calculated as a percentage of the “permeation amount” compared to the “applied amount”, and it is known that the permeation amount of chemical substances becomes constant under infinite dose conditions [48].

Thus, if the dermal absorption test is performed under conditions of a sufficiently high application amount (=infinite dose) for other drugs, the permeation amount becomes constant, and only the amount of application can be increased. So, dermal absorption rate is expected to decrease. Frasch et al. [46] also suggested that skin absorption rate was highly influenced by applied load of chemicals.

This study was expected to be helpful in terms of regulation of pharmaceuticals and various chemical substances as well as cosmetics. However, this test had a limitation that only two materials were used. It seemed to be helpful in this study to compare various results using test substances with various LogKow values. Also, increasing the number of repetitions for this study would be helpful. It would help reduce data distribution and improve data consistency.

## Figures and Tables

**Figure 1 pharmaceutics-13-00641-f001:**
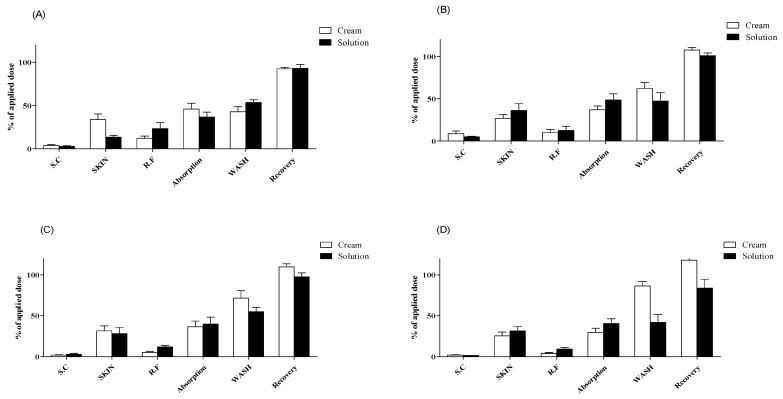
Dermal absorption of caffeine in rat skins (*n* = 4, mean ± SD). Dermal absorption test results with doses of (**A**) 5 mg/cm^2^ or 5 µL/cm^2^, (**B**) 10 mg/cm^2^ or 10 µL/cm^2^, (**C**) 25 mg/cm^2^ or 25 µL/cm^2^, and (**D**) 50 mg/cm^2^ or 50 µL/cm^2^. SC, stratum corneum; SKIN, dermis and epidermis; RF, receptor fluid; WASH, remaining formulation rinsed using alcohol swabs.

**Figure 2 pharmaceutics-13-00641-f002:**
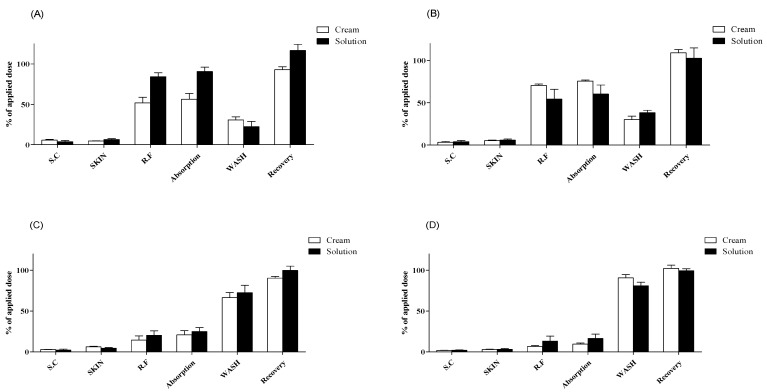
Dermal absorption of caffeine in Micropig^®^ skins (*n* = 4, mean ± SD). Dermal absorption test results with doses of (**A**) 5 mg/cm^2^ or 5 µL/cm^2^, (**B**) 10 mg/cm^2^ or 10 µL/cm^2^, (**C**) 25 mg/cm^2^ or 25 µL/cm^2^, and (**D**) 50 mg/cm^2^ or 50 µL/cm^2^. SC, stratum corneum; SKIN, dermis and epidermis; RF, receptor fluid; WASH, remaining formulation rinsed using alcohol swabs.

**Figure 3 pharmaceutics-13-00641-f003:**
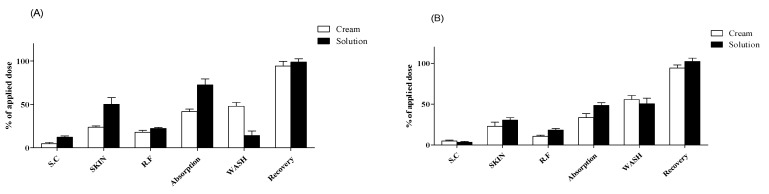
Dermal absorption of testosterone in rat skins (n = 4, mean ± SD). Dermal absorption test results with doses of (**A**) 5 mg/cm^2^ or 5 µL/cm^2^, (**B**) 10 mg/cm^2^ or 10 µL/cm^2^, (**C**) 25 mg/cm^2^ or 25 µL/cm^2^, and (**D**) 50 mg/cm^2^ or 50 µL/cm^2^. SC, stratum corneum; SKIN, dermis and epidermis; RF, receptor fluid; WASH, remaining formulation rinsed using alcohol swabs.

**Figure 4 pharmaceutics-13-00641-f004:**
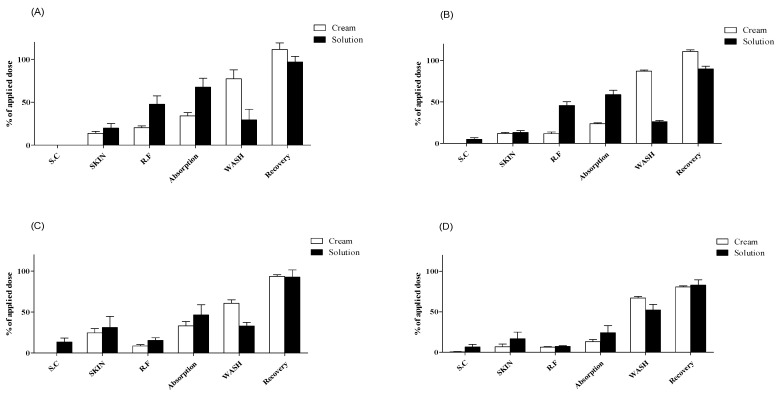
Dermal absorption of testosterone in Micropig^®^ skins (n = 4, mean ± SD). Dermal absorption test results with applied doses of (**A**) 5 mg/cm^2^ or 5 µL/cm^2^, (**B**) 10 mg/cm^2^ or 10 µL/cm^2^, (**C**) 25 mg/cm^2^ or 25 µL/cm^2^, and (**D**) 50 mg/cm^2^ or 50 µL/cm^2^. SC, stratum corneum; SKIN, dermis and epidermis; RF, receptor fluid; WASH, remaining formulation rinsed using alcohol swabs.

**Figure 5 pharmaceutics-13-00641-f005:**
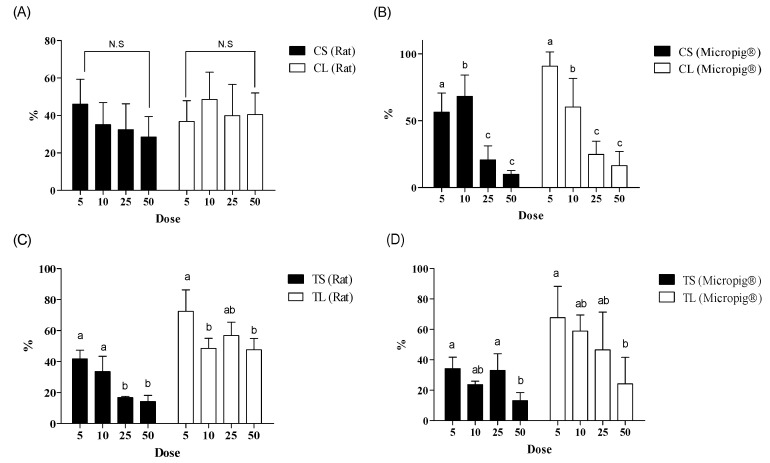
Comparing the dermal absorption rate (%, mean ± SD) of caffeine and testosterone, depending on the dose (5, 10, 25, and 50 mg/cm^2^ or μL/cm^2^). (**A**) Caffeine (rat), (**B**) caffeine (Micropig^®^), (**C**) testosterone (rat), and (**D**) testosterone (Micropig^®^). The same letter being written on the bar graph means no statistically significant difference in each data item (ANOVA with Tukey’s test; *p* = 0.05). CS, caffeine in cream formulation; CL, caffeine in solution; TS, testosterone in cream formulation; TL, testosterone in solution formulation; ANOVA, analysis of variance.

**Figure 6 pharmaceutics-13-00641-f006:**
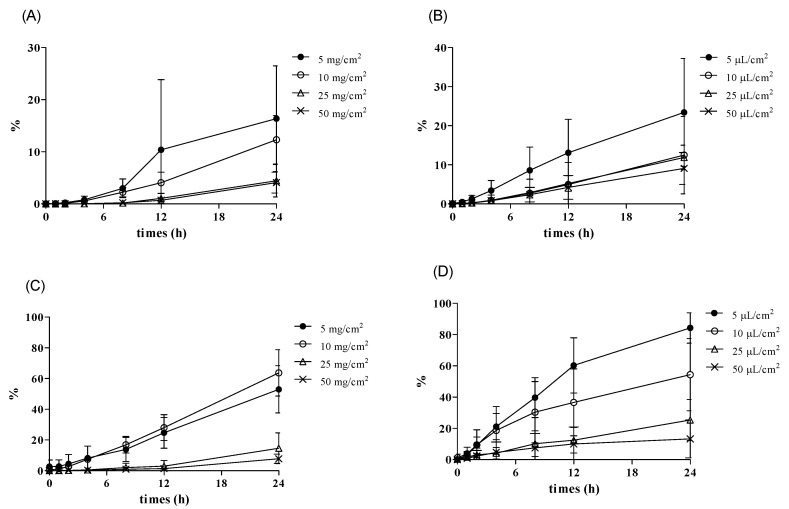
Penetration of caffeine (5, 10, 25, 50 mg or μL/cm^2^) (mean ± SD). (**A**) Rat (cream), (**B**) rat (solution), (**C**) Micropig^®^ (cream), and (**D**) Micropig^®^ (solution).

**Figure 7 pharmaceutics-13-00641-f007:**
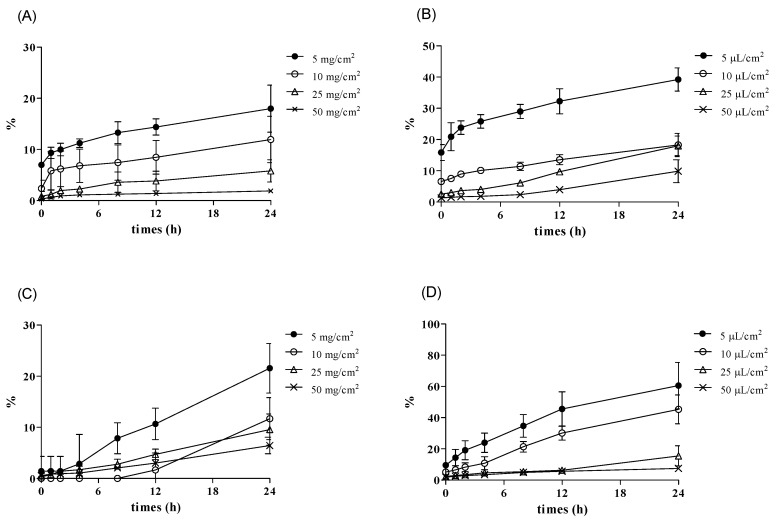
Penetration of testosterone (5, 10, 25, 50 mg or μL/cm^2^) (mean ± SD). (**A**) Rat (cream), (**B**) Rat (solution), (**C**) Micropig^®^ (cream), and (**D**) Micropig^®^ (solution).

**Table 1 pharmaceutics-13-00641-t001:** Physicochemical properties of caffeine and testosterone.

Properties	Caffeine	Testosterone
Chemical structure	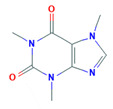	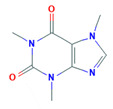
INCI name	Caffeine	Testosterone
IUPAC name	1,3,7-trimethylpurine-2,6-dione	(8R,9S,10R,13S,14S,17S)-17-hydroxy-10,13-dimethyl-1,2,6,7,8,9,11,12,14,15,16,17-dodecahydrocyclopenta[a]phenanthren-3-one
CAS Number	58-08-2	58-22-0
EC Number	200-362-1	200-370-5
Molecular formula	C_8_H_10_N_4_O_2_	C_19_H_28_O_2_
Molecular weight	194.19 g/mol	288.4 g/mol
Log Kow	−0.07	3.32
Solubility	In water, 2.16 × 10^4^ mg/L at 25 °C	In water, 23.4 mg/L at 25 °C

## Data Availability

Not applicable.

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
