# Peer review of "Effect of Application Amounts on In Vitro Dermal Absorption Test Using Caffeine and Testosterone"

_pharmaceutics, 2021, doi:10.3390/pharmaceutics13050641_

Round 1

Reviewer 1 Report

The main question of the paper is centered on measuring if dermal absorption rate would be significantly different when following OECD guidelines and SCCS PROTOCOLS in Franz cell diffusion chamber for dermal absorption when varying: amounts and drug dosage forms (1% caffeine and 0,1% testosterone - cream and solution) on different animal skins (minipigs and rats).

The paper is easy to follow and does have a logical flow with sufficient details. The main weakness is the lack of novelty and the discussion of results. So the authors should explore/discuss in more detail all data presented. For example, Figures 6 and 7 (Penetration data) were not even cited in the text.

Try to comment on why using a UV-VIs method instead of HPLC stability-indicating method as it may be more suitable when degradation products may be expected for a long experimental protocol as seen here?

Minor changes include revision of the text such as: 

  • Fig 3C does not show the same absorption rate as cited on the text (page 13, 2nd paragraph, 3rd line: "....significantly decreased to 14.2-16.8% in cream formulation"
  • page 12: preposition to or for

Author Response

The main question of the paper is centered on measuring if dermal absorption rate would be significantly different when following OECD guidelines and SCCS PROTOCOLS in Franz cell diffusion chamber for dermal absorption when varying: amounts and drug dosage forms (1% caffeine and 0,1% testosterone - cream and solution) on different animal skins (minipigs and rats).The paper is easy to follow and does have a logical flow with sufficient details. The main weakness is the lack of novelty and the discussion of results. So the authors should explore/discuss in more detail all data presented. For example, Figures 6 and 7 (Penetration data) were not even cited in the text.

→ I added description about Fig. 6-7 in p.13, 3rd – 5th paragraph (Discussion).

We could also show that this dose range is suitable for obtaining dermal absorption rate based on the kinetics of dermal permeability. In most of the dermal permeation kinetics graphs (Fig. 6-7) for the two test substances, the skin absorption rate (the slope of the graph) gradually decreased in the test under the application dose condition of 5 and 10 mg/cm² or 25 and 50 μL/cm², and it meant that these doses were the "finite dose” conditions used to calculate the skin absorption rate.

However, some exceptions were appeared in the case using Micropig skin and a cream formulaltion, the rate of skin absorption (the slope of the graph) tended to increase gradually (Fig. 6C, 7C). Therefore, it was difficult to define 5 and 10 mg/cm2 as a “finite dose” under this condition.

Nevertheless, it showed a sufficiently high skin absorption rate at 5 and 10 mg/cm2, and as mentioned above, the skin absorption rate was similar to that of human in vivo skin absorption rate. Therefore, it was judged to be still suitable for the test.

Try to comment on why using a UV-VIs method instead of HPLC stability-indicating method as it may be more suitable when degradation products may be expected for a long experimental protocol as seen here?

→ We could not observed that testosterone was naturally degraded during the analysis period. But the UV-Vis detector is simple to operate, inexpensive, and has a variety of references, so we used it. There is another method using HPLC with UV-Vis detector (https://www.sigmaaldrich.com/technical-documents/articles/analytical-applications/hplc/hplc-analysis-of-testosterone-and-1-testosterone-g005413.html).

Also, we validated this method, and it resulted in high accuracy, precision and specificity, so we determined that it was suitable.

Please consider it.

Minor changes include revision of the text such as: 

Fig 3C does not show the same absorption rate as cited on the text (page 13, 2nd paragraph, 3rd line: "...significantly decreased to 14.2-16.8% in cream formulation"

→ Thanks for it. It is a typo. It is Fig. 5C not Fig. 3C.

page 12: preposition to or for

→ We do not understand the comment. Sorry about it.

Reviewer 2 Report

Some revisions are needed:

  • Page 2, section 2. Formulations: The authors should indicate the type of cream vehicle (hydrophilic or lipophilic) and its components, considering that even the authors specified in the Introduction section that the vehicle (lipophilicity) is one of the factors that affect the dermal absorption of a product.
  • Page 3, section 3. Skin preparation: Please indicate the body region of the Minipig from which the skin membrane was taken.
  • Page 3, section 4. In vitro Franz diffusion cell equipment: Please indicate the pH value of PBS solution and if sink conditions were achieved using the specified receptor media.
  • Page 3, section 4. In vitro Franz diffusion cell equipment: “A skin area of 1.77 cm² was exposed, and 5, 10, 25, and 50 mg/cm² or 5, 10, 25, and 50 μL/cm² of cream or liquid formulation, respectively, containing 1% caffeine or 0.1% testosterone was applied on the exposed area.” >>> The authors should explain how they managed to apply such small amounts of formulations on the specified skin area, so the test conditions were accomplished. Generally, for this test area, the donor compartment must be filled with aprox. 300 mg formulation, in order to assure the appropriate test conditions.
  • Page 10, section Discussion: “The reason for selecting the liquid vehicle as an ethanol solution was to dissolve the solid test sub-stance, so it was inevitable.” >>> Please rephrase.
  • Page 10, section Discussion: “The skin used in the test was 500 um Micropig…” >>> please correct the measure unit.
  • Page 10, section Discussion: “The skin used in the test was 500 um Micropig, and the full skin of rats was used. This was used because the TEWL values (7.4 ± 0.5 g/h/m2, Micropig; 13.5 ± 1.3 g/h/m2, Rat) were most similar to men’s forearm global TEWL standard (7.7 ± 2 g/h/m2) as an indicator of the integrity of the skin in the test…”>>> The authors should justify more clearly the choice of skin membranes of different thicknesses, considering TEWL values.

Author Response

Page 2, section 2. Formulations: The authors should indicate the type of cream vehicle (hydrophilic or lipophilic) and its components, considering that even the authors specified in the Introduction section that the vehicle (lipophilicity) is one of the factors that affect the dermal absorption of a product.

→ I added description about composition of cream formulation in p.3, 1st  paragraph (2.2 Formulations).

The cream formulations were made by Cosmecca Korea Inc (Eumsung, Korea). It contained 1% caffeine or 0.1% testosterone. These were made using a mixture of several chemicals (sorbitan stearate, PEG-100 stearate, glyceryl stearate, cetearyl alcohol, polysorbate 60, hydrogenated polydecene, caprylic/capric triglyceride, dimethicone, disodium EDTA, chlorphenesin, glycerin, propanediol, distilled water, ammonium acryloyldimethyltaurate/VP copolymer, polyisobutene, polysorbate 20, polyacrylate-13, sorbitan isostearate, phenoxyethanol, caprylyl glycol and ethylhexylglycerin).

Page 3, section 3. Skin preparation: Please indicate the body region of the Minipig from which the skin membrane was taken.

→ I revised the sentence of the third line from the bottom in Page 3, section 3.

Dorsal skin membranes were obtained from 6-month-old minipigs and standardized for dermal absorption experiments (thickness 500 μm; size 3 × 3 cm²). Each skin membrane was stored frozen at –20℃ until further use.

Page 3, section 4. In vitro Franz diffusion cell equipment: Please indicate the pH value of PBS solution and if sink conditions were achieved using the specified receptor media.

→ I added information of the pH value of PBS in the 5th line, Page 3, section 4.

The receptor chamber was filled with PBS (pH 7.4) and 6% (v/v) PEG in PBS as an RF for caffeine and testosterone, respectively. Skin membranes were thawed to room temperature, loaded onto the Franz diffusion cell,

Page 3, section 4. In vitro Franz diffusion cell equipment: “A skin area of 1.77 cm² was exposed, and 5, 10, 25, and 50 mg/cm² or 5, 10, 25, and 50 μL/cm² of cream or liquid formulation, respectively, containing 1% caffeine or 0.1% testosterone was applied on the exposed area.” >>> The authors should explain how they managed to apply such small amounts of formulations on the specified skin area, so the test conditions were accomplished. Generally, for this test area, the donor compartment must be filled with aprox. 300 mg formulation, in order to assure the appropriate test conditions.

→ I added this sentence in 12th line, Page 3, section 4.

When applied for this test area, the donor compartment must be filled with approx. 8.85, 17.7, 44.25 and 88.5 mg formulation using sterile syringes, in order to assure the appropriate test conditions.

Page 10, section Discussion: “The reason for selecting the liquid vehicle as an ethanol solution was to dissolve the solid test substance, so it was inevitable.” >>> Please rephrase.

→ I rephrased this sentence in 10th line, Page 10 (Discussion).

The reason for choosing the liquid vehicle as an ethanol solution was to dissolve the test substance, so it was inevitable to use an organic solution, even if it could affect the skin condition.

Page 10, section Discussion: “The skin used in the test was 500 um Micropig…” >>> please correct the measure unit.

→ I corrected the unit (um → µm)

Page 10, section Discussion: “The skin used in the test was 500 um Micropig, and the full skin of rats was used. This was used because the TEWL values (7.4 ± 0.5 g/h/m2, Micropig; 13.5 ± 1.3 g/h/m2, Rat) were most similar to men’s forearm global TEWL standard (7.7 ± 2 g/h/m2) as an indicator of the integrity of the skin in the test…”>>> The authors should justify more clearly the choice of skin membranes of different thicknesses, considering TEWL values.

→ I added a sentence to further justify the argument.

The TEWL value is a factor that describes the properties of the skin barrier, and the skin barrier has a great influence on the rate of skin absorption (Elkeeb et al, 2010). Since we wanted to use the skin that best mimics the human skin barrier, we chose the thickness condition for each skin with similar TEWL values. In addition, the optimum thickness with stable TEWL value has been selected.

Ref: Elkeeb, R., Hui, X., Chan, H., Tian, L., & Maibach, H. I. (2010). Correlation of transepidermal water loss with skin barrier properties in vitro: comparison of three evaporimeters. Skin Res Technol. 2010 Feb;16(1):9-15.

Reviewer 3 Report

This manuscript reports the effects of application doses of caffeine and testosterone for the in vitro dermal absorption test. The authors found that dermal absorption of both compounds decreased with increasing amounts of application and concluded that a low dose of the formulation should be applied for risk exposure assessment. The overall study was well organized, and the results are clearly presented to support the conclusion. There are suggestions to improve the manuscript.

1. The selection of caffeine and testosterone as model compounds needs to be justified in Introduction section.

2. If the finding of the decreased dermal absorption with increasing application doses could be applied for other drugs, it needs to be discussed.

Author Response

This manuscript reports the effects of application doses of caffeine and testosterone for the in vitro dermal absorption test. The authors found that dermal absorption of both compounds decreased with increasing amounts of application and concluded that a low dose of the formulation should be applied for risk exposure assessment. The overall study was well organized, and the results are clearly presented to support the conclusion. There are suggestions to improve the manuscript.

  1. The selection of caffeine and testosterone as model compounds needs to be justified in Introduction section.

→ I added description about the reason in 3rd paragraph in page 1 (Introduction).

This study was performed using caffeine and testosterone as test materials. These were used as reference chemicals for skin absorption tests recommended by OECD TG 428. These chemicals had different physicochemical properties; LogKow values for caffeine and testosterone are –0.07 and 3.32, respectively, and molecular weights are 194.19 and 288.4 g/mol, respectively. In addition, these chemicals had a variety of skin absorption data (Lehman et al., 2011; Bronaugh and Franz., 1986; Dreher et al., 2002; Sandt et al., 2004; Kamel et al., 2008; Korting et al., 2008; Trauer et al., 2009; Trauer et al., 2010; Lehman et al, 2010; Lehman and Raney., 2012; Veryser et al., 2013; Guth et al., 2015; Wester and Maibach., 1986).

  1. If the finding of the decreased dermal absorption with increasing application doses could be applied for other drugs, it needs to be discussed.

→ Yes, I added the description about it in 3rd line from the bottom, p.14 (Discussion).

It was expected that other drugs will have a similar trend to the results of this study. In general, the dermal absorption rate is calculated as a percentage of the “permeation amount” compared to the “applied amount”, and it is known that the permeation amount of chemical substances becomes constant under infinite dose conditions (Lau &. Ng, 2017).

Thus, if the dermal absorption test is performed under conditions of a sufficiently high application amount (= infinite dose) for other drugs, the permeation amount become constant, and only the amount of application can be increased. So, dermal absorption rate expected to decrease. Frasch et al. (2014) also suggested that skin absorption rate was highly influenced by applied load of chemicals.

Round 2

Reviewer 1 Report

The author answered the questions satisfactorily, except for the comments transcripted below.  

We could not observe that testosterone was naturally degraded during the analysis period. But the UV-Vis detector is simple to operate, inexpensive, and has a variety of references, so we used it. There is another method using HPLC with UV-Vis detector (https://www.sigmaaldrich.com/technical-documents/articles/analytical-applications/hplc/hplc-analysis-of-testosterone-and-1-testosterone-g005413.html).

Could the authors show additional evidence of no sample degradation during the experiment? You may upload chromatograms of the samples and a summary of the validated parameters as supplementary material. Also, some reference(s) work(s) used for optimizing the analytical methods for caffeine and testosterone would be great. 

Author Response

Could the authors show additional evidence of no sample degradation during the experiment? You may upload chromatograms of the samples and a summary of the validated parameters as supplementary material. Also, some reference(s) work(s) used for optimizing the analytical methods for caffeine and testosterone would be great.

→ Thank you for the comment. According to reviewer’s comment, the evidence of no sample degradation is shown in the supplementary Tables showing validation parameters of both caffeine and testosterone. Related references are also inserted. Please consider it.